# Think Smarter, Focus Wisely: Adaptive Cognitive Allocation for LLM Reasoning

## Abstract

Encouraging longer chains of thought is a common design practice for improving LLM reasoning. However, recent studies show that "thinking more" can backfire. In response, prior studies have typically employed augmented reasoning strategies to enhance performance. While these approaches often improve reasoning robustness and yield higher accuracy, they may also generate excessively long chains, which introduce redundant checks, demand disproportionate reasoning effort, and ultimately lead to inefficient consumption of cognitive resources. This paper introduces **A**daptive **R**easoning via **C**ognitive **A**llocation (**ARCA**), a structured reasoning framework that adaptively allocates cognitive resources across reasoning phases based on their reasoning state, thereby mitigating the efficiency–accuracy trade-off. The core idea of ARCA is to structure the reasoning procedure into classified phases, while grounding the process and suppressing incoherent drift. Within each phase, ARCA generates candidate directions and employs a Borda-Aggregated selector to identify the most promising ones, while steering inference along phase-aware directions and pruning redundant exploration. Through the dynamic allocation of cognitive resources, the proposed ARCA framework can achieve a balance between accuracy and efficiency. Across six reasoning benchmarks, ARCA consistently outperforms strong baselines, either in terms of enhanced accuracy or reduced reasoning cost.

## 1 Introduction

"More thinking should mean better answers." This intuition feels natural to humans and has heavily influenced the design of large language models (LLMs). The prevailing wisdom suggests that encouraging models to generate longer, more detailed chains of thought is a reliable path to stronger reasoning (Wei et al., 2022; OpenAI, 2025; 2024). While LLMs guided by this principle have shown remarkable capabilities in complex tasks like question answering (Lewkowycz et al., 2022), knowledge retrieval (Schmidgall et al., 2025) and decision support systems (Lubos et al., 2025), a troubling paradox emerges: pushing them to simply "think more" can backfire. Recent studies (Jin et al., 2025) reveal that excessive reasoning, or "overthinking," often degrades performance. Extended chains of reasoning may accumulate errors (Lewkowycz et al., 2022), reduce stability (Wang et al., 2023), and increase computational demand—leading to substantially higher inference costs and limiting their practical usability.

To counteract this fragility, a dominant strategy has been to enhance robustness and achieve higher accuracy by introducing augmented reasoning procedures that supplement the original inference process. For instance, methods like pairwise comparison (Zhang et al., 2024) and iterative self-evaluation (Chen et al., 2024b) reduce errors by generating and assessing multiple solution paths, but this incurs massive computational overhead. Other techniques, such as those relying on pre-defined skill libraries (Chen et al., 2024a), can impose structural rigidities that limit their adaptability to novel problems. In essence, these methods achieve robustness at such a high cost in inference demand and inflexibility that they become impractical for many real-world applications.

On the other side, efficiency-oriented methods aim to reduce inference costs, typically through streamlined meta-reasoning architectures (Sui et al., 2025b; Patil & Jadon, 2025). However, this efficiency is achieved by relying on handcrafted contextual frameworks and human-defined heuristics, thereby specializing the systems for specific tasks. This specialization fundamentally limits their

problem-solving scope and leads to poor generalizability. Consequently, a generalizable framework that allows models to adaptively allocate their reasoning effort or we called it **cognitive resources** to achieve both accuracy and efficiency remains a critical, underexplored challenge. The central question therefore becomes: *How can LLMs learn to allocate their cognitive resources adaptively to achieve both accuracy and efficiency simultaneously?*

The key insight is that **effective reasoning requires selective focus rather than indiscriminate depth**. Humans intuitively follow this principle. Consider a game of Sudoku shown in Figure 1: a player instantly fills in a number when it is the only possibility in a row—an act of efficient, linear deduction. However, when confronted with a complex intersection of constraints, the same player may pencil in multiple candidates in a few cells, exploring their implications before committing. This represents an on-demand expansion of the reasoning process to ensure the next move is robust. We posit that endowing LLMs with a similar capability for dynamic cognitive resource allocation is key to resolving the tension between accuracy and efficiency in reasoning. To achieve this, **LLMs likewise need to learn to allocate their cognitive resources adaptively**: engaging in deep reasoning when necessary and pruning effort when a path proves unpromising.

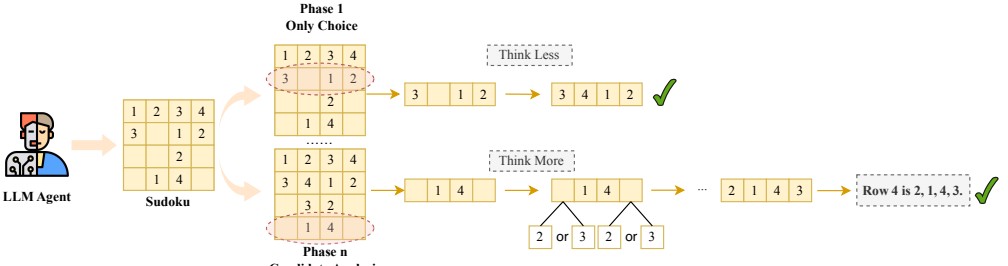

Figure 1: LLMs operate through multiple phases to solve a task, with certain phases requiring minimal cognitive resources while others demand in-depth reasoning.

To realize adaptive cognitive resource allocation in LLMs, we propose a structured reasoning framework called **A**daptive **R**easoning via **C**ognitive **A**llocation (**ARCA**). ARCA guides LLMs in deciding where and how much to think during reasoning by operating through two synergistic stages: **reasoning chain construction** and **cognitive resource allocation**. In the reasoning chain construction stage, a *phase generator* decomposes complex tasks into logically ordered phases, specifying the goal of each step. This structured decomposition prevents fragmented reasoning and ensures systematic task coverage. To illustrate, let us return to the Sudoku example introduced earlier. The solving process can be structured into phases such as [*Fill cells with unique candidates, Explore candidate intersections, Validate placements, and others*], which serve as clear anchors guiding reasoning toward the solution.

Building on this structure, the *cognitive resource allocation* stage dynamically manages reasoning effort at two levels. At the macro level, a phase classifier directs LLMs focus to the current phase's objectives, ensuring efficient progress. At the micro level, a direction generator explores candidate reasoning steps, while a proposed *Borda-Aggregated selection mechanism* guided by LLMs' preference feedback to choose the most promising path. ARCA enables the LLM to adaptively allocate more cognitive resources to critical phases of a task, thereby facilitating precise reasoning while preserving overall processing efficiency. Crucially, the structured phases from the first stage provide the precise context needed for the second stage to make informed allocation decisions, ensuring that cognitive resources are invested exactly where they are most needed. Extensive experiments on six diverse reasoning tasks demonstrate that ARCA achieves strong performance while maintaining favorable resource efficiency compared to baseline approaches.

Main contributions of this work are concluded as: 1) The problem of efficient cognitive resource allocation is formalized, the critical balance between accuracy and efficiency addresses a core challenge in the reasoning of LLMs; 2) An LLM reasoning framework ARCA is proposed to realize cognitive resource allocation in LLM reasoning, where a high-level task decomposition guides fine-grained adaptive exploration to simultaneously improve both reasoning accuracy and efficiency; 3) Comprehensive experiments on diverse reasoning tasks demonstrate that our method achieves strong performance while maintaining favorable resource efficiency compared to baseline approaches.

## 2 RELATED WORK

**Reasoning Methods for Robustness**   A dominant strategy to enhance reasoning robustness is to generate and evaluate a multitude of augmented reasoning procedures. This approach moves beyond a single chain of thought (Wei et al., 2022) by explicitly constructing multiple reasoning trajectories. For instance, Tree of Thoughts (Yao et al., 2023) frames reasoning as a tree search, allowing exploration of parallel thought candidates at each step. Graph of Thoughts (Besta et al., 2024) further generalizes this into a graph structure to capture more complex interdependencies between thoughts. Methods like Boosting Task-Oriented Reasoning (Chen et al., 2024b) iteratively generate numerous reasoning steps and use LLM-based error analysis to refine them, while Comparative Tree of Thought (Zhang et al., 2024) employs pairwise comparison to select optimal paths. CoDT (Wang et al., 2025) improves the robustness against reference corruption by providing a few exemplars with structured and defensive reasoning as demonstrations. MME-CoT (Jiang et al., 2025) incorporates three novel metrics to assess the reasoning quality, robustness, and efficiency. Math-RoB (Yu et al., 2025b) uses an instruction-based approach to generate diverse datasets resembling training distributions. CD-CoT (Zhou et al., 2024) enhances LLMs' denoising-reasoning capabilities by contrasting noisy rationales with one clean rationale. CoT-GCG (Su, 2024) enhances adversarial attacks on aligned LLMs by integrating CoT prompts with the greedy coordinate gradient technique.

**Efficiency-Oriented Reasoning Methods**   In direct contrast, another line of research focuses on streamlining the reasoning process to reduce computational costs. These methods often employ meta-reasoning architectures or probabilistic approximations to achieve faster inference. For example, Meta-Reasoner (Sui et al., 2025b) uses contextual multi-armed bandits to dynamically adjust reasoning strategies based on state evaluation. ES-CoT (Mao et al., 2025) shortens thought generation by prompting the LLM to output a step answer at each reasoning step. THINK-Bench (Li et al., 2025c) introduces a benchmark with novel efficiency metrics, to evaluate the reasoning efficiency. (Cui et al., 2025) proposes a method that identifies and focuses on generating important reasoning steps in reasoning by using perplexity to measure their importance, (Sui et al., 2025a) explores efficient data use, small language model reasoning, and evaluation methods. Soft Chain-of-Thought (Xu et al., 2025) leverages probabilistic soft chains and prompt tweaks for efficient, uncertainty-aware reasoning. Similarly, COAT (Shen et al., 2025) uses action-oriented chains for meta-reasoning without full model tuning.

**Feedback-Based Refinement and Evaluation**   Another influential line of research focuses on iterative self-improvement through feedback mechanisms. In this paradigm, the LLM itself is leveraged to evaluate and refine its reasoning trajectories in a cyclic manner. For instance, Self-Refine (Madaan et al., 2023) introduces an algorithm where the LLM generates output, provides self-feedback, and then refines its output based on that feedback. RCO (Yu et al., 2025a) trains critic models using a feedback loop where critiques guide the actor model in refining responses. RefCritic (Tang et al., 2025) trains a critic module with dual rule-based rewards focusing on instance-level correctness of solution judgments and refinement accuracies of the policy model. (Renze & Guven, 2024) investigates the effects of self-reflection in large language models on problem-solving performance. (Potamitis & Arora, 2025) enhances reasoning by allowing the models to retry problem-solving attempts upon identifying incorrect answers. RFM-RAG (Li et al., 2025a) transforms stateless retrieval into stateful continuous knowledge management by constructing a dynamic evidence pool to generate refined queries using relational triples and evidence. PHP (Zheng et al., 2023) incorporates the solution from a previous attempt as a hint for the next, creating an iterative improvement loop.

While significant progress has been made within these methods, the fundamental tension between robustness and efficiency remains largely unaddressed. ARCA addresses this core challenge by empowering LLMs to dynamically allocate greater cognitive resources to critical phases of a reasoning task, achieving an optimal balance between accuracy and efficiency.

## 3 ADAPTIVE REASONING VIA COGNITIVE ALLOCATION

In this section, we present a comprehensive introduction to the ARCA framework, which achieves cognitive resource allocation to balance accuracy and efficiency in LLM reasoning. It integrates two complementary components: reasoning chain construction, which provides structural guidance

to mitigate fragmented reasoning, and cognitive resource allocation, which enables selective and adaptive distribution of cognitive effort across phases and reasoning steps. Through the interaction of these two components, ARCA enables LLMs to concentrate their reasoning effort on the phases most critical, thus **improving accuracy** and **enhancing efficiency**.

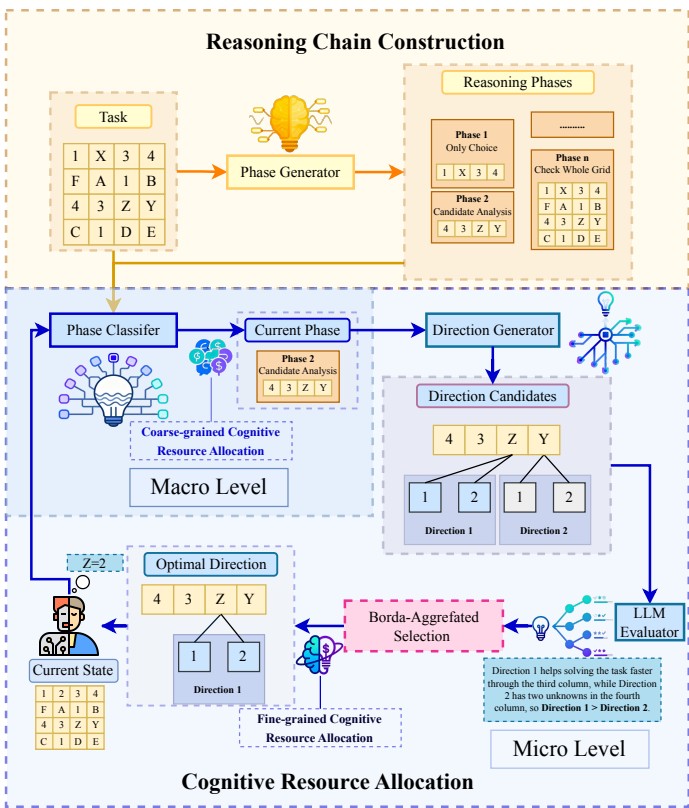

Figure 2: The overall framework of ARCA. It integrates reasoning chain construction and cognitive resource allocation to balance accuracy and efficiency in LLM reasoning, where reasoning chain construction structures the task into phases, and cognitive resource allocation allocates cognitive resources through macro-level phase guidance and micro-level direction selection.

## 3.1 REASONING CHAIN CONSTRUCTION

We first introduce the *Reasoning Chain Construction*, a module that leverages the semantic analysis capability of LLMs to build a high-level blueprint and to define the reasoning phase (allocation units) that serve as the basis for cognitive resource allocation in ARCA. Traditional chain-of-thought prompting often produces unstructured, divergent, or incomplete reasoning trajectories (Ji et al., 2024). Without explicit semantic scaffolding, the model may digress into irrelevant steps or overlook crucial components of the task. To achieve ARCA's goal of balancing accuracy and efficiency, it is essential to first identify the reasoning phases where resources can be meaningfully allocated.

For illustration, consider solving a Sudoku: the task can be decomposed into several reasoning phases, each serving as an allocation unit. A natural sequence begins with (i)*Single-candidate placement*, filling cells with only one valid option; followed by (ii) *Candidate elimination and inference*, iteratively deducing placements based on row, column, and block constraints; (iii) *Conflict detection and backtracking*, revising decisions if contradictions arise; and (iv) *Validation and finalization*, ensuring all cells satisfy Sudoku rules.

To realize this decomposition, ARCA introduces the *Reasoning Phase Generator (ReasonGen)*, which maps a task $\mathcal{T}$ into a sequence of logically ordered reasoning phases:

$$\mathcal{P} = \{\phi_1, \phi_2, \ldots, \phi_n\} = \text{ReasonGen}(\mathcal{T}), \tag{1}$$

where $\mathcal{T}$ is the input task, $\phi_i$ is the $i$-th reasoning phase and $\mathcal{P}$ is the generated phase set. ReasonGen begins by analyzing the task context to identify and delineate each fundamental phase. Each phase specifies a semantic objective—*what needs to be achieved* at that step rather than prescribing the operational details of *how* to achieve it. For the Sudoku example, this naturally yields phases starting with single-candidate placement, followed by iterative candidate elimination, conflict detection and backtracking, and concluding with final validation. We provide illustrative examples of the generated phases in Appendix B.4. This structured decomposition defines the allocation units (reasoning phases) required by ARCA and constrains reasoning within explicit semantic boundaries. By providing a high-level blueprint for cognitive resource allocation, ReasonGen reduces irrelevant exploration and yields more coherent and efficient inference compared to unstructured CoT.

## 3.2 Cognitive Resource Allocation

Building on the reasoning phases and high-level blueprint defined by Reasoning Chain Construction, this module performs the allocation of cognitive resources used by ARCA. Allocating resources uniformly across all reasoning steps is inefficient and costly; ARCA instead dynamically adjusts cognitive resource allocation according to the current semantic phase and task context. The allocation process uses two coordinated mechanisms: **Macro-Level Phase Identification** and **Micro-Level Direction Selection**. The former determines *which phase* the reasoning process is currently in, while the latter identifies the most promising direction within that phase.

At the **Macro-Level**, the module prioritizes the current phase, biasing resource allocation toward operations relevant to it. To identify the current phase, we introduce a phase classifier *PhaseClass*, a runtime supervisory unit that dynamically evaluates the solver's state to assign the reasoning step to the appropriate phase:

$$\phi_t = \text{PhaseClass}(x_t, \mathcal{P}), \tag{2}$$

where $x_t$ is the reasoning state at step $t$. The architecture enhances complex reasoning by dynamically identifying the current phase, enabling the LLM to focus on phase-specific objectives. Once a phase is completed, the module transitions seamlessly to the next, reallocating computational resources according to new requirements. By concentrating effort on the active phase and minimizing expenditure on completed or irrelevant directions, the system maintains targeted and efficient progress across the reasoning chain, thereby preventing resource over-allocation. **A key insight** underlying this design is that reasoning phases naturally impose heterogeneous resource demands: simpler phases require minimal resources and benefit from rapid closure, whereas more demanding phases call for deeper inference and thus greater resource investment. The architecture capitalizes on this heterogeneity, allowing adaptive allocation of computational effort so that resources are concentrated on ongoing objectives while avoiding over-allocation to irrelevant directions. In doing so, the system achieves dynamic and efficient resource utilization throughout reasoning.

At the **Micro-Level**, the module generates and selects reasoning directions that are locally optimal or highly relevant to the active phase. Cognitive resources or reasoning efforts are dynamically assigned to the most promising next steps within the phase. To this end, we introduce a reasoning direction generator *DirectionGen*, a real-time strategic module that steers the LLM's inference process:

$$\mathcal{D}_t = \{d_1, d_2, \ldots, d_m\} = \text{DirectionGen}(\phi_t, x_t, \mathcal{T}), \tag{3}$$

where $\mathcal{D}_t$ is the set of candidate directions at step $t$ and $d_i$ is the $i$-th direction. At each reasoning step, the generator produces a focused set of actionable directions based on the current phase and contextual state. These directions provide timely, targeted guidance aligned with the phase's objectives. To enable broad exploration, multiple candidate paths are proposed. **Selecting the most promising path** among them poses a central challenge, as explicit reward signals are unavailable and handcrafted reward functions are difficult to design and prone to bias. To address this, we employ an LLM as an implicit preference oracle to evaluate candidate directions.

Building on these evaluations, we introduce a **Borda-aggregated direction selection algorithm** (Yan et al., 2022), which consolidates pairwise comparisons into a robust consensus score and will be described in detail in section 3.3. This approach mitigates noise from individual judgments, reduces reliance on brittle heuristics, and reliably identifies the most promising reasoning direction. The optimal direction $d_t^*$ is then selected according to the aggregated Borda score:

$$d_t^* = \underset{d \in \mathcal{D}_t}{\arg\max} \, \text{Borda}(d). \tag{4}$$

The LLM then generates the next reasoning state $x_{t+1}$ according to:

$$x_{t+1} = \text{LLM}(x_t, \phi_t, d_t^*). \tag{5}$$

By providing such fine-grained tactical guidance, this module effectively bridges phase planning with real-time reasoning, enabling efficient and rational allocation of cognitive resources throughout the inference process. In the next subsection, we will make a detailed introduction to the Borda-aggregated direction selection algorithm.

### 3.3 BORDA-AGGREGATED DIRECTION SELECTION

In this section, we provide a detailed explanation of how the Borda-aggregated direction selection algorithm identifies the most preferred direction in Equation 4. To select the optimal reasoning path via LLM-based preference comparisons, prior work has often adopted the dueling bandit framework (Zhang et al., 2024). In this setting, when comparing two candidate thoughts $i$ and $j$, candidate $i$ is chosen with probability $q(i, j)$, while candidate $j$ is selected with the complementary probability $q(j, i) = 1 - q(i, j)$. Here, $q(i, j) \geq \frac{1}{2}$ whenever $i$ is ranked higher than $j$. Repeated comparisons are assumed to be independent.

However, dueling bandit algorithms such as DTS (Wu & Liu, 2016) typically rely on the Copeland score (Zoghi et al., 2015) to aggregate comparison outcomes. A major limitation of the Copeland score in LLM-based preference assessment is its sensitivity to minor preference variations (Goel et al., 2017). This sensitivity arises from its win-counting mechanism, which can amplify stochastic fluctuations inherent in LLM judgments (Li et al., 2025b). Consequently, achieving stable rankings often requires extensive comparisons, which is especially challenging in noisy evaluation environments (Qin et al., 2023). To address this limitation, the Borda score (Rothe, 2019) is adopted as an alternative. The Borda score for a candidate direction $i$ is defined as:

$$\text{Borda}(i) = \frac{1}{|\mathcal{C}| - 1} \sum_{j \in \mathcal{C}, j \neq i} q(i, j),$$

where $\mathcal{C}$ denotes the set of candidates. The Borda score's win-rate formulation effectively aggregates pairwise preferences and offers clear practical advantages in LLM evaluation settings. Its scoring mechanism, which estimates the average probability of victory, is well-suited to the stochastic and noisy nature of LLM judgments. By averaging outcomes across multiple comparisons, it confers robustness against minor inconsistencies in individual assessments (Rothe, 2019). Furthermore, the computational simplicity of maintaining and updating win rates enables highly efficient implementation in large-scale scenarios, allowing broad candidate coverage without exhaustive evaluations.

We formulate direction selection as a Borda score-based framework (Yan et al., 2022; Clarke et al., 2021), with an LLM serving as the **preference function**. Our algorithm begins with a pruning phase to efficiently eliminate clearly suboptimal directions while retaining the most promising candidates. During each pruning iteration, approximate Borda scores are computed by comparing each candidate against a fixed-size random subset of opponents. This sparse comparison strategy ensures broad coverage without exhaustive evaluations. Candidates with scores below a elimination score are pruned. We set the elimination score at 0.5, which corresponds to random chance performance, while any candidate scoring below this level is deemed inferior and removed. This pruning process is repeated iteratively until the number of remaining candidates falls below a predefined threshold. The algorithm then proceeds to a final evaluation stage, conducting full round-robin comparisons among the remaining candidates. This enables accurate, high-confidence estimation of the true Borda scores, from which the top-scoring candidate is chosen as the final solution. By combining efficient broad pruning with precise final assessment, this two-stage approach effectively balances computational efficiency with selection reliability. Details are provided in Appendix C.

## 4 EXPERIMENTS

In this section, we conduct comprehensive experiments to evaluate performance and provide an in-depth analysis of ARCA, comparing it against baseline approaches in terms of cost and accuracy. Additional results and ablation study are in Appendix B.2 and B.3.

Table 1: Performance comparison on different datasets

| Methods | Datasets | | | | | Average (%) |
|---|---|---|---|---|---|---|
| | AQUA (%) | BBEH (%) | GSM8K (%) | Game of 24 (%) | AIME (%) | |
| CoT | 69.2 | 9.1 | 70.9 | 65.7 | 51.4 | 53.3 |
| Self-Refine | 75.3 | 8.9 | 71.4 | 73.5 | 70.3 | 59.9 |
| SToT | 72.2 | 7.3 | 69.2 | 61.3 | 62.1 | 54.4 |
| PoT | 66.8 | 12.9 | 92.4 | 86.8 | 65.5 | 64.9 |
| CToT | 85.4 | 28.2 | 84.8 | 76.3 | **74.1** | 69.8 |
| BoT | 79.3 | 20.2 | 93.8 | 73.1 | 46.1 | 62.5 |
| **ARCA** | **86.2** | **54.5** | **96.7** | **91.3** | 72.7 | **80.3** |

Table 2: Average accuracy on Sudoku Puzzles

| Method | Acc. 3×3 (%) | Acc. 4×4 (%) | Acc. 5×5 (%) |
|---|---|---|---|
| CoT | 90.0 | 73.3 | 60.0 |
| Self-Refine | 83.3 | 90.0 | 50.0 |
| SToT | 100.0 | 80.0 | 30.0 |
| PoT | 90.0 | 70.0 | 60.0 |
| CToT | 100.0 | 100.0 | 76.3 |
| BoT | 90.0 | 90.0 | 70.0 |
| **ARCA** | **100.0** | **100.0** | **90.0** |

## 4.1 REASONING TASKS

We evaluate the performance of our proposed method, ARCA, on a suite of six challenging real-world reasoning tasks. These tasks span a diverse range of domains, including question answering (AQUA), multi-step arithmetic (BBEH), math word problems (GSM8K), the Game of 24, Sudoku puzzles, and the AIME competition-level problems.

- **AQUA** (Wei et al., 2022), the question answering task, which consists of 254 arithmetic reasoning questions designed to evaluate logical reasoning abilities through diverse mathematical problems. Each question is associated with five multiple-choice options labeled A through E.

- **BBEH** (Kazemi et al., 2025) is a recently introduced benchmark aimed at advancing the evaluation of reasoning in large language models. It replaces each original task in BBH (Suzgun et al., 2022) with a novel variant that targets comparable reasoning skills while substantially increasing the difficulty. In our experiments, we select the multi-step arithmetic task from BBEH. This task incorporates new arithmetic operators, some of which are defined in terms of other operators. It also introduces a compositional operation format.

- **GSM8K** (Cobbe et al., 2021) is a widely-used benchmark of grade-school math word problems that require multi-step reasoning to solve. Each problem involves basic arithmetic operations and logical thinking to arrive at the final answer. The dataset contains high-quality linguistically diverse questions, making it a standard testbed for evaluating the mathematical reasoning capabilities.

- **The Game of 24** (Yao et al., 2023) is a mathematical challenge in which the objective is to combine four given numbers using basic arithmetic operations to yield a total of 24. For our experiments, we adopt the same dataset and setup as, which includes 1,362 problems sourced from 4nums.com.

- **The Sudoku** (Long, 2023) includes 10 puzzles each for 3×3, 4×4, and 5×5 grid sizes. Each puzzle is partially filled, and the task is to complete the grid without altering the provided numbers. A solution is considered correct if the completed grid adheres to all standard Sudoku rules.

- **AIME** (Mathematical Association of America, 2024) is a highly challenging mathematics contest administered to top-performing participants of the AMC. It serves as a key benchmark for evaluating the mathematical reasoning and problem-solving capabilities of LLMs.

All experiments were conducted with the `Deepseek-V3` model (DeepSeek-AI, 2025). Detailed configurations and results, including those with other models, are provided in Appendix B.

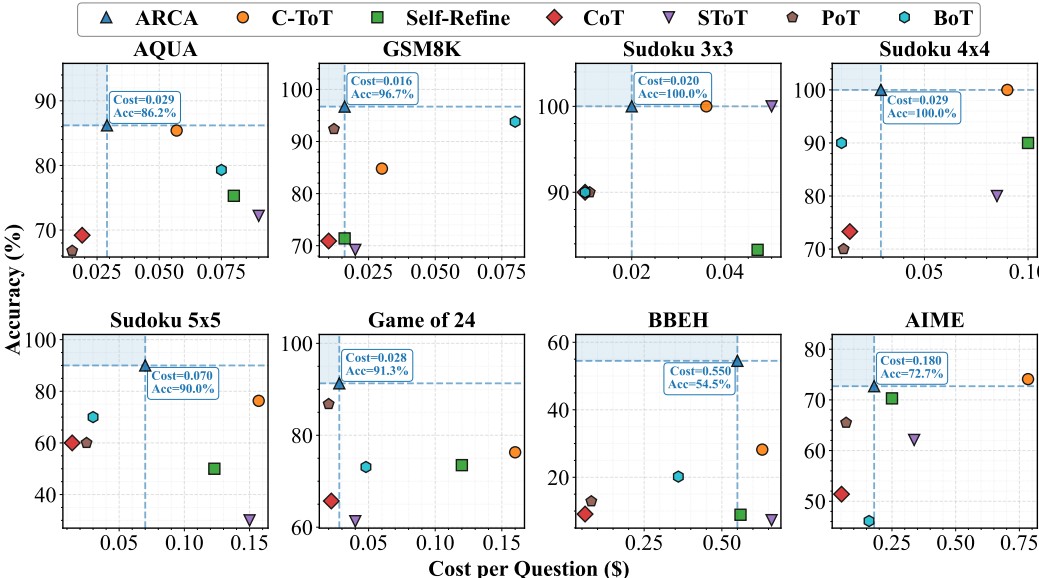

Figure 3: Accuracy–cost comparison across reasoning tasks. Each marker denotes a method. Dashed lines and the text box highlight ARCA, which none surpass in the shaded region, showing ARCA performs best among compared methods.

## 4.2 BASELINES AND RESULTS

We compare our method with the following baselines: CoT (Wei et al., 2022), Self-Refine (Madaan et al., 2023), SToT (Yao et al., 2023), PoT (Chen et al., 2023), CToT (Zhang et al., 2024) and BoT (Chen et al., 2024b). On each dataset, 3 test runs are conducted and the average accuracy as well as the cost per question are presented. The experimental results are presented in Table. 1 and Table. 2. These results demonstrate that ARCA outperforms other baselines and achieves significant advantages, particularly in complex tasks such as the Game of 24, Sudoku puzzles, and BBEH.

In these tasks, the solver requires long-chain reasoning and operates in a high-dimensional solution space. The heuristic strategies for reasoning in ARCA provide critical guidance at each step, assisting the solver in accurately steering toward the final answer, reducing deviations, and ultimately leading to more effective and reliable problem-solving. This mechanism proves essential for navigating the complexity inherent in such challenging domains.

To evaluate the efficiency of cognitive resource allocation, we employ reasoning cost as a key metric, where using fewer reasoning costs indicates more efficient. A comparison of the accuracy and cost between our method and baseline approaches across different tasks is presented in Fig. 3. Here we adopt the cost calculation method from CToT (Zhang et al., 2024). The experimental findings clearly illustrate that ARCA successfully achieves an effective and practical balance between model accuracy and operational cost-efficiency. This balance is realized through the novel integration of structured reasoning chain construction and dynamic cognitive resource allocation mechanisms. Our method demonstrates the capability to autonomously identify and prioritize critical reasoning phases, thereby allocating computational resources in an adaptive and context-aware manner. This sophisticated mechanism not only consistently enhances the quality and reliability of solutions but also maintains coherent focus throughout the reasoning trajectory toward the correct solution. Consequently, ARCA delivers substantially improved overall reasoning performance while simultaneously ensuring judicious control over computational expenditure, providing a reliable framework for complex cognitive tasks.

## 4.3 ABLATION STUDY

**Analysis on the parameters in Borda-aggregated direction selection.** To analyze how the the parameters in the direction selection algorithm affects accuracy and computational cost, we conduct

experiments on the GSM8K dataset. To ensure comparability, the maximum number of direction is fixed at 12 across all trials. Experiments are conducted with pruning pool size threshold $m = \{4, 6, 8, 12\}$ (*where $m = 12$ corresponds to the non-pruning case*) and the number of comparisons per direction during pruning set to $n = \{2, 4, 6\}$. Results in Table. 3 indicate that when using pool size pruning, configurations with pool size threshold 6 and 8 achieve performance comparable to the full pool (size 12), while significantly reducing computational expense. Although the value of $n$ has an impact on performance, experiments show that a medium $n$ is sufficient to achieve rapid pruning in the early stages. This demonstrates that the proposed selection framework effectively balances computational efficiency with selection reliability, thus offers a scalable and practical solution for resource-aware automated reasoning.

**Analysis on the Maximum Direction Number.** The number of generated directions determines the breadth of exploration available to the solver at each reasoning step. We evaluate the impact of this parameter by conducting experiments on the AQUA and Game of 24 datasets, using maximum direction numbers set to 2, 4, 6, and 10. The results, summarized in Table. 4, indicate that when the maximum direction number is limited to 2 which resulting in a narrow exploration scope, the performance is noticeably worse compared to configurations allowing broader exploration. In contrast, when the maximum number of directions is set to 4 or higher, performance stabilizes and remains consistently high. This suggests that the reasoning chain construction mechanism helps the solver maintain a clear objective and reduces the need for extensive exploration, thereby achieving more efficient and reliable problem-solving even with moderate search width.

Table 3: Ablation study on direction selection algorithm parameters on GSM8K. Here $n$ denotes the number of comparisons and $m$ denotes the size. **Acc.** is accuracy (%), and **Cost** is average computational cost (lower is better).

|       | $m = 4$ | | $m = 6$ | | $m = 8$ | | $m = 12$ | |
|-------|------|------|------|------|------|------|------|------|
|       | Acc. | Cost | Acc. | Cost | Acc. | Cost | Acc. | Cost |
| $n = 2$ | 89.2 | 0.040 | 95.2 | 0.045 | 94.5 | 0.053 | 95.8 | 0.096 |
| $n = 4$ | 92.9 | 0.058 | 96.1 | 0.064 | 96.4 | 0.075 | 95.8 | 0.096 |
| $n = 6$ | 94.1 | 0.075 | 96.3 | 0.080 | 96.3 | 0.082 | 95.8 | 0.096 |

Table 4: Ablation study on the number of directions $d_n$. Accuracy (%) is reported on AQUA and Game of 24, along with the average. Larger $d_n$ generally improves performance, with diminishing gains after $d_n = 6$.

| Direction Number $d_n$ | AQUA (%) | Game of 24 | Average (%) |
|-------|------|------|------|
| $d_n = 2$ | 79.5 | 88.5 | 84.0 |
| $d_n = 4$ | 85.4 | 91.3 | 88.4 |
| $d_n = 6$ | 86.2 | 91.3 | 88.7 |
| $d_n = 10$ | 85.7 | 93.1 | 89.4 |

## 5 CONCLUSION

In conclusion, we propose a novel framework ARCA designed to tackle efficient cognitive resource allocation in LLM reasoning, with a specific focus on balancing accuracy and efficiency. By integrating decomposition, strategy generation, monitoring, and dynamic selection into a cohesive system, our approach enhances structural coherence, optimizes reasoning effort, and improves accuracy in complex scenarios while keeping additional reasoning cost negligible to preserve efficiency. Extensive experiments across diverse tasks show that our method delivers strong performance while maintaining competitive resource efficiency compared to existing baselines. In future work, we will pursue more reliable reasoning chains and refine our framework for accurate direction generation, focusing on more complex task environments.

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

## A  USE OF LARGE LANGUAGE MODELS (LLMs)

Our study investigates the reasoning capabilities of large language models (LLMs). Accordingly, all experiments were conducted on LLMs to evaluate and validate our proposed approach. Beyond experimentation, we employed an LLM as an auxiliary tool during manuscript preparation. Specifically, it was used to refine language for grammar and clarity, and to generate illustrative (non-experimental) figures based on prompts we provided. All research ideas, methods, experiments, analyses, and conclusions were developed by the authors.

## B  EXPERIMENTS AND SETTINGS

### B.1  THE DETAILS OF EXPERIMENT

We evaluate the performance of our proposed method, ARCA, on a suite of six challenging real-world reasoning tasks. These tasks span a diverse range of domains, including question answering (AQUA), multi-step arithmetic (BBEH), math word problems (GSM8K), the Game of 24, Sudoku puzzles, and the AIME competition-level problems.

**AQUA** (Wei et al., 2022), the question answering task, which consists of 254 arithmetic reasoning questions designed to evaluate logical reasoning abilities through diverse mathematical problems. Each question is associated with five multiple-choice options labeled A through E. In this experiment, we set the pruning pool size threshold $m = 3$, the number of comparisons per direction during pruning phase $n = 3$, max generated directions to 6, max depth of reasoning to 3.

**BBEH** (Kazemi et al., 2025) is a recently introduced benchmark aimed at advancing the evaluation of reasoning in large language models. It replaces each original task in BBH (Suzgun et al., 2022) with a novel variant that targets comparable reasoning skills while substantially increasing the difficulty.In our experiments, we select the multi-step arithmetic task from BBEH. This task incorporates new arithmetic operators, some of which are defined in terms of other operators. It also introduces a compositional operation format. In this experiment, we set the pruning pool size threshold $m = 4$, the number of comparisons per direction during pruning phase $n = 4$, max generated directions to 8, max depth of reasoning to 6.

**GSM8K** (Cobbe et al., 2021) is a widely-used benchmark of grade-school math word problems that require multi-step reasoning to solve. Each problem involves basic arithmetic operations and logical thinking to arrive at the final answer. The dataset contains high-quality linguistically diverse questions, making it a standard testbed for evaluating the mathematical reasoning capabilities. In this experiment, we set the pruning pool size threshold $m = 3$, the number of comparisons per direction during pruning phase $n = 3$, max generated directions to 6, max depth of reasoning to 3.

**The Game of 24** (Yao et al., 2023) is a mathematical challenge in which the objective is to combine four given numbers using basic arithmetic operations to yield a total of 24. For our experiments, we adopt the same dataset and setup as, which includes 1,362 problems sourced from 4nums.com. In this experiment, we set the pruning pool size threshold $m = 3$, the number of comparisons per direction during pruning phase $n = 3$, max generated directions to 6, max depth of reasoning to 6.

**The Sudoku** (Long, 2023) includes 10 puzzles each for 3×3, 4×4, and 5×5 grid sizes. Each puzzle is partially filled, and the task is to complete the grid without altering the provided numbers. A solution is considered correct if the completed grid adheres to all standard Sudoku rules. In this experiment, we set the pruning pool size threshold $m = 4$, the number of comparisons per direction during pruning phase $n = 4$, max generated directions to 8, max depth of reasoning to 6.

**AIME** (Mathematical Association of America, 2024) is a highly prestigious and challenging mathematics contest administered to top-performing participants of the AMC. It serves as a key benchmark for evaluating the mathematical reasoning and problem-solving capabilities of large language models. Here we set the pruning pool size threshold $m = 4$, the number of comparisons per direction during pruning phase $n = 4$, max generated directions to 8, max depth of reasoning to 6.

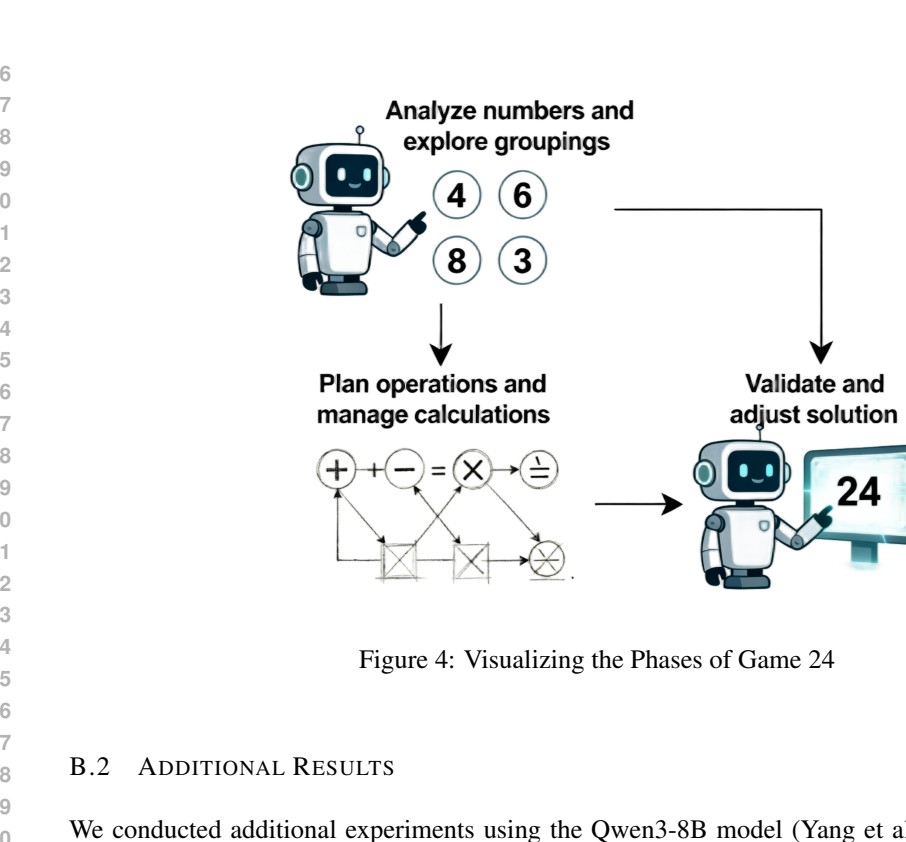

Figure 4: Visualizing the Phases of Game 24

## B.2 ADDITIONAL RESULTS

We conducted additional experiments using the Qwen3-8B model (Yang et al., 2025), comparing our results with two key baselines: the fundamental CoT method and one of the top-performing baselines, CToT. The results are presented in Table 6. We also include experimental results on the smaller model, Qwen2.5-7B and the proprietary GPT-4o-mini, in Table 7.

## B.3 ADDITIONAL ABLATION STUDY

To evaluate our method, we conducted ablation studies on the AQUA, GSM8K, and Game of 24 datasets Table 5, focusing on two key questions: 1) whether the selection is sensitive to the initial ordering of options, and 2) how critical the Borda count component is. Specifically, we designed two experimental variants: **Random**, where we shuffle the options before each Borda count to eliminate positional bias; **w/o Gen**, where the phase generator is removed from our framework; and **w/o Borda**, where we replace the Borda count with a simpler selection process to isolate its contribution.

The experimental results demonstrate that our Borda selection method effectively mitigates positional bias, maintaining robust performance even when the option order is randomized. Meanwhile, the phase generator helps guide the direction of reasoning and plays an important role in complex reasoning tasks. Furthermore, the component ablation study confirms the critical role of the Borda framework within the selection pipeline. It significantly reduces selection biases of the LLM while ensuring algorithmic stability, thereby enabling efficient and accurate selection of subsequent reasoning paths.

Table 5: Additional ablation study

| Methods | AQUA (%) | GSM8K(%) | Game of 24(%) |
|---|---|---|---|
| Random | 84.8 | 95.9 | 91.8 |
| w/o Gen | 72.4 | 70.6 | 69.9 |
| w/o Borda | 76.1 | 79.4 | 75.2 |
| ARCA | 86.2 | 96.7 | 91.3 |

### B.4 THE ANALYSIS OF THE GENERATED PHASES

In this section, we evaluate the quality of the generated phases, illustrated with concrete examples from Sudoku, Game of 24, and GSM8K. We also provide a visual representation of the phases for Game of 24 in Figure 4.

**Phases of Sudoku** ['*Basic Elimination', 'Candidate Reasoning', 'Guessing and Backtracking*'].

**Phases of Game 24** ['*Analyze numbers and explore groupings - Identify relationships and test pairing possibilities', 'Plan operations and manage calculations - Determine sequence and ensure mathematical viability', 'Validate and adjust solution - Verify results and refine approach to achieve 24*'].

**Phases of GSM8K** ['*Calculate the total number of eggs consumed for breakfast and baking', 'Calculate the daily earnings from selling the remaining eggs', 'Final verification or solution step and give the final answer*'].

While these phases are often broader and exhibit less consistency compared to those created by human experts, they prove to be sufficiently accurate to effectively guide the reasoning process.

Moreover, the system demonstrates a degree of fault tolerance: even coarse or imperfect phase decompositions rarely lead to catastrophic failures. This robustness is achieved because the downstream Phase Classifier and Borda-based selection mechanism work in concert to steer the model toward phase-relevant reasoning trajectories, effectively compensating for upstream imperfections.

Table 6: Additional results with Qwen3-8B

| Methods | Datasets | | | | | Average (%) |
|---|---|---|---|---|---|---|
| | AQUA (%) | BBEH (%) | GSM8K (%) | Game of 24 (%) | Sudoku Puzzle 5×5 (%) | |
| CoT | 79.4 | 16.7 | 79.2 | 75.7 | 70.0 | 64.2 |
| CToT | 84.9 | 28.8 | 87.7 | 82.4 | 73.3 | 71.4 |
| ARCA | 90.6 | 47.1 | 93.5 | 91.1 | 90.0 | 82.4 |

Table 7: Additional results with Qwen2.5-7B and GPT-4o-mini

| Models | Datasets | | | | |
|---|---|---|---|---|---|
| | AQUA (%) | BBEH (%) | GSM8K (%) | Game of 24 (%) | Sudoku Puzzle 5×5 (%) |
| Qwen2.5-7B | 84.7 | 40.2 | 89.4 | 90.2 | 80 |
| GPT-4o-mini | 89.8 | 45.6 | 94.7 | 88.3 | 90 |

## C IMPLEMENTATION DETAILS

### C.1 DETAILS OF THE SELECTION ALGORITHM

In this section, we provide a detailed explanation of how the Borda-aggregated direction selection algorithm identifies the most preferred direction in Equation 4. To select the optimal reasoning path via LLM-based preference comparisons, prior work has often adopted the dueling bandit framework (Zhang et al., 2024). In this setting, when comparing two candidate thoughts $i$ and $j$, candidate $i$ is chosen with probability $q(i, j)$, while candidate $j$ is selected with the complementary probability $q(j, i) = 1 - q(i, j)$. Here, $q(i, j) \geq \frac{1}{2}$ whenever $i$ is ranked higher than $j$.

However, dueling bandit algorithms such as DTS (Wu & Liu, 2016) typically rely on the Copeland score (Zoghi et al., 2015) to aggregate comparison outcomes. A major limitation of the Copeland score in LLM-based preference assessment is its sensitivity to minor preference variations (Goel et al., 2017). This sensitivity arises from its win-counting mechanism, which can amplify stochastic fluctuations inherent in LLM judgments (Li et al., 2025b). Consequently, achieving stable rankings

often requires extensive comparisons, which is especially challenging in noisy evaluation environments (Qin et al., 2023). To address this limitation, the Borda score (Rothe, 2019) is adopted as an alternative. The Borda score for a candidate direction $i$ is defined as:

$$\text{Borda}(i) = \frac{1}{|\mathcal{C}| - 1} \sum_{\substack{j \in \mathcal{C} \\ j \neq i}} q(i, j),$$

where $\mathcal{C}$ denotes the set of candidates. The Borda score's win-rate formulation effectively aggregates pairwise preferences and offers clear practical advantages in LLM evaluation settings. Its scoring mechanism, which estimates the average probability of victory, is well-suited to the stochastic and noisy nature of LLM judgments. By averaging outcomes across multiple comparisons, it confers robustness against minor inconsistencies in individual assessments (Rothe, 2019). Furthermore, the computational simplicity of maintaining and updating win rates enables highly efficient implementation in large-scale scenarios, allowing broad candidate coverage without exhaustive evaluations.

We formulate direction selection as a Borda score-based framework (Yan et al., 2022; Clarke et al., 2021), with an LLM serving as the preference function. In practice, we define the $q(i, j)$ as a binary indicator:

$$q(i, j) = \begin{cases} 1 & \text{if direction } i \text{ is preferred over direction } j \\ 0 & \text{otherwise} \end{cases}$$

Our algorithm begins with a pruning phase to efficiently eliminate clearly suboptimal directions while retaining the most promising candidates. During each pruning iteration, approximate Borda scores are computed by comparing each candidate against a fixed-size random subset of opponents. This sparse comparison strategy ensures broad coverage without exhaustive evaluations. Candidates with scores below a elimination score are pruned. We set the elimination score at 0.5, which corresponds to random chance performance, while any candidate scoring below this level is deemed inferior and removed. This pruning process is repeated iteratively until the number of remaining candidates falls below a predefined threshold. In **extreme cases** where the scores of all candidates are close to 0.5, making them difficult to distinguish quickly, the pruning phase will be halted after a limited number of rounds if no clear selection has been made. The system then proceeds by selecting the top-m directions with the highest current scores and enters the final phase.

The algorithm then proceeds to a final evaluation stage, conducting full round-robin comparisons among the remaining candidates. This enables accurate, high-confidence estimation of the true Borda scores, from which the top-scoring candidate is chosen as the final solution. By combining efficient broad pruning with precise final assessment, this two-stage approach effectively balances computational efficiency with selection reliability. Details and further analysis are provided in Algorithm 1 and Appendix C.4.

**Complexity Analysis** We measure cost by the number of LLM preference queries `preference(u, v)`. Let the initial pool size be $K_0$ and the pruning threshold be $m$. In each pruning iteration, every direction is compared against $n$ sampled opponents, yielding $\mathcal{O}(Kn)$ queries when the current pool size is $K$. Let $K_t$ denote the pool size at iteration $t$ until it reaches $K_T \leq m$, giving total pruning cost $\sum_{t=1}^{T} K_t n$. In the best case, the pool shrinks geometrically (e.g., removing a constant fraction each iteration), so $\sum_t K_t = \mathcal{O}(K_0)$ and the pruning cost is $\mathcal{O}(nK_0)$. In the worst case, the pool shrinks only by $\mathcal{O}(1)$ items per iteration, giving $T = \mathcal{O}(K_0)$ and $\sum_t K_t = \mathcal{O}(K_0^2)$, for a pruning cost of $\mathcal{O}(nK_0^2)$. After pruning, the final pool of size $K' \leq m$ undergoes full pairwise comparison, costing $\mathcal{O}(K'^2) \leq \mathcal{O}(m^2)$. Therefore the overall complexity is $\mathcal{O}(nK_0 + m^2)$ in the best case and $\mathcal{O}(nK_0^2 + m^2)$ in the worst case; under simplifications and approximations, the dominant terms are $\mathcal{O}(nK_0)$ and $\mathcal{O}(nK_0^2)$ respectively.

## C.2 THE PROMPT EXAMPLE OF DIFFERENT COMPONENTS

**Reasoning Chain Generator** The reasoning chain generator in our framework is designed using the following prompt. It begins by performing a detailed task analysis to identify and delineate fundamental reasoning phases. It emphasizes logical continuity between phase, and each phase clearly defines what should be achieved, rather than prescribing how. Based on this analysis, it constructs a clear and efficient reasoning blueprint. This blueprint directs subsequent operations along

logically coherent pathways and prioritizes high-value reasoning trajectories. As a result, the generator ensures comprehensive problem coverage, significantly improves computational efficiency, and overcomes the limitations of unstructured chain-of-thought reasoning.

```
chain_prompt = '''You are an expert task decomposer. Your role is
    to analyze complex problems and break them down into essential
     high-level sub-phases. Each sub-phase should represent a
    critical milestone that moves toward solving the task.
```

---

**Algorithm 1** Borda-Aggregated Direction Selection

---

1: **Input:**
2: Pool: Set of $K$ reasoning directions $\{1, \ldots, K\}$ to select
3: $n$: Number of comparisons per direction during pruning phase
4: $m$: Final pool size threshold
5: Preference$(u, v)$: LLM comparison function

6: **while** $K > m$ **do**                 ▷ **Pruning Phase**
7:   $E \leftarrow \emptyset$
8:   **for** each direction $i \in \{1, \ldots, K\}$ in the Pool **do**
9:    samples $\leftarrow$ randomly select $\min(n, K-1)$ directions from $\{1, \ldots, K\} \setminus \{i\}$
10:    **for** each $j \in$ samples **do**
11:     $E \leftarrow E \cup \{(i, j)\}$
12:    **end for**
13:   **end for**
14:   $\mathbf{W} \leftarrow \mathbf{0}^K, \mathbf{C} \leftarrow \mathbf{0}^K$              ▷ Reset counters
15:   **for** each $(u, v) \in E$ **do**
16:    winner $\leftarrow$ preference$(u, v)$
17:    $\mathbf{C}[u] \leftarrow \mathbf{C}[u] + 1, \mathbf{C}[v] \leftarrow \mathbf{C}[v] + 1$
18:    $\mathbf{W}[\text{winner}] \leftarrow \mathbf{W}[\text{winner}] + 1$
19:   **end for**
20:   Borda $\leftarrow [\mathbf{W}[i]/\mathbf{C}[i] \text{ for } i \in \{1, \ldots, K\}]$
21:   NewPool $\leftarrow \{i \mid \text{Borda}[i] \geq 0.5\}$
22:   Pool $\leftarrow$ NewPool, $K \leftarrow |\text{Pool}|$
23: **end while**

24: $E_{\text{final}} \leftarrow \{(i, j) \mid i, j \in \text{Pool}, i \neq j\}$        ▷ **Final Evaluation Phase**
25: $\mathbf{W} \leftarrow \mathbf{0}^K, \mathbf{C} \leftarrow \mathbf{0}^K$
26: **for** each $(u, v) \in E_{\text{final}}$ **do**
27:   winner $\leftarrow$ preference$(u, v)$
28:   $\mathbf{C}[u] \leftarrow \mathbf{C}[u] + 1, \mathbf{C}[v] \leftarrow \mathbf{C}[v] + 1$
29:   $\mathbf{W}[\text{winner}] \leftarrow \mathbf{W}[\text{winner}] + 1$
30: **end for**
31: Borda $\leftarrow [\mathbf{W}[i]/\mathbf{C}[i] \text{ for } i \in \{1, \ldots, K\}]$

32: **return** $\{j \mid \text{Borda}[j] = \max_{i \in \{1, \ldots, K\}} \text{Borda}[i]\}$    ▷ Set of all max-scoring directions

---

```
**** Generate only the most essential sub-phases needed to
    complete this task, excluding all implementation details and
    optional steps.
**SUB-Phase DEFINITION:**
Each sub-phase should specify WHAT needs to be accomplished, not
    HOW to do it. Focus on the key objectives that must be
    achieved.

**Example Demonstrations:**

    **Geometry Problem:**
```

```
    Task: "Find the area of a triangle with base 8cm and height 5cm
        "

    Key Sub-Phases:
    1. Identify the area formula for triangles ,
    2. Extract given dimensions from the problem ,
    3. Compute the area using the formula
    4. Final verification or solution step and give the final
        answer

**Output Format Strictly Follow This Pattern:**
1. [Action-oriented sub-phase description] ,
2. [Next essential sub-phase] ,
...

**Critical Reminders:**
- Phases should answer "what needs to be done" not "how to do it"
- Avoid transitional words ("then", "next", "after")
- Exclude mathematical symbols, formulas, or specific methods
- Maintain consistent verb tense and clarity
- Ensure sub-phases are truly sequential and complementary
'''
```

**Phase Classifier**   The phase classifier enhances complex reasoning by dynamically identifying the current phase in real time, enabling the solver to strategically allocate computational resources toward phase-specific objectives. Once a phase concludes, the module seamlessly transitions to another, directing the LLM's resources to the most relevant ongoing stage. By focusing efforts on the active phase and reducing investments in completed or irrelevant directions, it maintains efficient and targeted progress throughout the reasoning process, thereby avoiding wasteful allocation.

```
classifier_prompt = f'''You are a Sub-phase Reasoning Engine. I
    will give the sub-phase list:{phases_list} and current
    thinking progress:{current_context}. Analyze the task progress
     and determine:
Which sub-phase should be actively worked on now.
***********
Choose the sub-phase from the list:{phases_list}, give me the
    number of index in the list.

**TASK ANALYSIS PROCESS:**
1. Compare current progress with each sub-phase's requirements
2. Identify the most immediate sub-phase that needs attention
3. Verify the selection matches logical progression

**OUTPUT FORMAT STRICTLY FOLLOW:**
[index]

**EXAMPLE:**
Phases: ["Data collection", "Analysis", "Validation"]
Current Context: "Finished gathering raw data, need to process it"
You choose [Analysis]
Output:
1

**CRITICAL RULES:**
- ******** Sub-phase MUST be from provided list and return in the
    number of index
- No explanations or additional text
- Sub-phase should logically follow from current_context
```

```
'''
```

**Reasoning Direction Generator**    At each reasoning step, the reasoning direction generator takes the current phase and contextual state as input and produces a focused set of actionable, executable directions. These outputs provide timely and targeted guidance aligned with the specific objectives of the phase.

```
direction_prompt = f'''You are a Phase-Oriented Direction
    Generator. Given the current step and the phase which needs to
     be achieved, Generate between {min_directions} and {
    max_directions} practical methods (directions) to achieve the
    specified phase.
**Current step: {current_step}
**Phase:** {current_phase}

**Direction Definition:**
Each direction should be a concrete, actionable method that:
1. Directly contributes to achieving the phase according to
    current step
2. Represents a distinct approach or technique
3. Is executable without external knowledge

**Output Requirements:**
- Generate between {min_directions} and {max_directions}
    directions
- Each direction must start with an action verb
- Format each direction as a bullet point ("- [direction
    description]")
- Keep directions concise (5-15 words)
- Exclude explanations or examples

**Quality Validation:**
- Each direction is a distinct method (not a restatement)
- Directions cover different aspects of the phase
- Methods are practical and executable
- Avoid overlapping or redundant directions

=== COMPLETE EXAMPLES ===
Example:
Current step: "Already generate several passwords."
Phase: "Validate password strength"
Directions:
- Check minimum password length
- Verify mixed character types
- Test against common passwords

=== FORMAT REQUIREMENTS ===
Output MUST be:
- [Direction 1]
- [Direction 2]
...
- [Direction n] (where n is between {min_directions} and {
    max_directions})
**Critical Rules:**
- STRICTLY use the given format
- NO numbering or other formats
- NO additional text outside bullet points
- Directions must answer "how to achieve the phase based on
    current step"
```

```
'''
```

**Thought Generator** We use the following prompt to generate the thought at each step, based on the task description, phase and optimal direction.

```
purpose_prompt = f'''You are a heuristic assistant specialized in
    sub-phase-based problem solving.

**CURRENT SUB-PHASE:** {phase}
**REQUIRED DIRECTION:** {direction[i]}

**TASK:** Generate exactly the next step that:
1. Directly applies the specified direction: "{direction[i]}"
2. Advances the current sub-phase: "{phase}"
3. Reach the phase as fast as possible !!!

**OUTPUT FORMAT RULES:**
- The next step should reach the phase as fast as possible.
- However, when the final step leads you to the final answer, give
     me only the numerical answer and print "###" before it,
    format as: ###[ANSWER]
- Otherwise, provide a clear action step
- No explanations, just the step itself

**VERIFICATION CHECKLIST:**
- Does this step directly follows the direction "{direction[i]}"?
- Does this step achieves the sub-phase "{phase}" as fast as
    possible?
- If final answer, does it start with "###"?
'''
```

**LLM comparison function** At each step, the generated direction are compared through Borda-aggregated direction selection framework using LLM comparison function, and the example of this comparison function is presented below.

```
preference_prompt='''As an analytical reasoning expert, critically
     evaluate which of the two reasoning paths demonstrates
    superior logical coherence, mathematical accuracy, and problem
    -solving effectiveness for the task. Consider: step-by-step
    validity, premise consistency, conclusion support, and error
    minimization. If both are objectively equal in all aspects,
    randomly select 1 or 2. Output must be exactly 1 or 2 with no
    additional text, explanations, or formatting.'''
```

### C.3 The Reasoning and Selection Process

We present the reasoning steps and selection process employed by ARCA on AIME and BBEH tasks. The results indicate that the reasoning procedure, along with the Borda selection mechanism, aligns closely with typical human decision-making patterns.

The reasoning and selection process of AIME task:

```
Step 0 (Round 0)
Phase: Select the 0th option (Establish the core geometric model)
Available options:
"Let speed be v, height h. At 45°: distance = h, at 60°: distance
    = h/√3. Distance covered in 10 min = h - h/√3 = h(1-1/√3)"
"Set up trigonometric equations: tan45° = h/x ⇒ h = x, tan60° = h/
    y ⇒ h = y√3"
```

```
"Complete solution: v×10 = h(1−1/√3̄), remaining distance h/√3̄,
    time = 10/(√3̄−1) = 5(√3̄+1)"
"Use cotangent: h cot45° = h, h cot60° = h/√3̄, find speed from 10
    min travel"
Selection: Option 2 – Set up basic trigonometric equations

Step 1 (Round 1)
Phase: Relate the key variables from different scenarios
Available options:
"Substitute values: x = h, y = h/√3̄, distance = h(1 − 1/√3̄)"
"Calculate distance traveled: x − y = y√3̄ − y = y(√3̄ − 1)"
"Equate height expressions: x = y√3̄"
"Introduce speed: v×10 = x − y"
Selection: Option 3 – Equate the two height expressions

Step 2 (Round 2)
Phase: Execute the complete algebraic derivation to solve for the
    target quantity
Available options:
"Calculate remaining time: t = y/v = y/[(x−y)/10] = 10y/(x−y) =
    10/(√3̄−1) = 5(√3̄+1)"
"Express distance traveled in 10 min as x − y"
"Rationalize denominator: 10/(√3̄−1) × (√3̄+1)/(√3̄+1) = 5(√3̄+1)"
"From x = y√3̄, find y = x/√3̄, then compute time"
Selection: Option 1 – Calculate the final remaining time with
    complete computation

Final Computation Process:
From x = y√3̄, distance in 10 min: x − y = y√3̄ − y = y(√3̄ − 1)
Speed v = y(√3̄ − 1)/10
Remaining distance to base: y
Time needed: t = y/v = y / [y(√3̄ − 1)/10] = 10/(√3̄ − 1)
Rationalize: t = 10(√3̄ + 1)/[(√3̄ − 1)(√3̄ + 1)] = 10(√3̄ + 1)/2 =
    5(√3̄ + 1)

Final Result
Algorithm answer: A [5(√3̄ + 1)]
Correct answer: A
```

The reasoning and selection process of BBEH task:

```
Step 0 (Round 0)
Phase: Establish the boundary framework for reasoning
Available options:
"Place the raven as the third bird from the left."
"Position robin immediately left of raven in the remaining slots (
    positions 2 and 3)."
"Determine the rightmost bird as quail."
"Identify the leftmost bird as owl: Position 1 = owl."
Selection: Option 4 – Identify the leftmost bird as owl: Position
    1 = owl.
Step 1 (Round 1)
Phase: Establish the key anchor point and determine the fixed
    point
Available options:
```

Table 8: Results of test case, modeling LLM preference feedback with uncertainty and ties.

| Algorithm | One found | Both found | Comparisons | Assessors |
|---|---|---|---|---|
| DTS(Copeland score) (Wu & Liu, 2016) | 710 | 7 | 1000 | 4–20 |
| Borda score method (Yan et al., 2022) | 733 | 81 | 616–795 | 3–6 |

```
"Assign positions: Position 1 = owl (leftmost), Position 5 = quail
    (rightmost)."
"Position robin left of raven: since raven is third, place robin
    in position 2."
"Given raven is third from left: Position 3 = raven. Robin is left
    of raven, so robin must be in position 2."
"Place the raven as the third bird from the left: Position 3 =
    raven."
Selection: Option 4 – Place the raven as the third bird from the
    left: Position 3 = raven.
Step 2 (Round 2)
Phase: Apply the remaining constraints to finalize the entire
    arrangement
Available options:
"Set quail at position 5 (rightmost)."
"Place the falcon in the only remaining position (position 4)."
"Position robin left of raven: since position 1 is owl and
    position 3 is raven, robin must be in position 2."
"Confirm all constraints are satisfied with the complete
    arrangement."
Selection: Option 1 – Set quail at position 5 (rightmost).
Final Computation Process:
Position 1: owl (given as leftmost)
Position 3: raven (given as third from left)
Position 2: robin (must be left of raven, and position 1 is
    occupied by owl)
Position 5: quail (given as rightmost)
Position 4: falcon (only remaining position)
The rightmost bird is in position 5, which is the quail.
Final Result:
Algorithm answer: A (The quail is the rightmost)
Correct answer: A
```

## C.4 ANALYSIS OF BORDA SCORE IN LLM FEEDBACK

To assess the ability of the Borda score to accommodate uncertainty, fine-grained distinctions, and potential ties commonly encountered in LLM preference feedback (Li et al., 2025b), we simulate a test scenario based on the setup and results from (Yan et al., 2022). The test case represents a scenario with no single winner and many ties, mirroring the challenges of LLM preference judgments.

**Test Case:**

$$q_{0,1} = q_{1,0} = 0.5$$
$$i > 1 \implies q_{0,i} = 0.75 \quad \text{and} \quad q_{i,0} = 0.25$$
$$i > 1 \implies q_{1,i} = 0.75 \quad \text{and} \quad q_{i,1} = 0.25$$
$$i > 1 \text{ and } j > 1 \implies q_{i,j} = 0.5$$

We adopted experimental parameters and evaluation criteria of (Yan et al., 2022). The experiment involved a large set of 100 options, and a fixed budget of 1000 comparisons. As shown in Table. 8, the simulation results demonstrate the superior performance of the Borda score method over the Copeland-based approach in identifying optimal outputs from LLM preference feedback.

