# OpenReview forum: "Think Smarter, Focus Wisely: Adaptive Cognitive Allocation for LLM Reasoning"
_ICLR.cc/2026/Conference — ICLR 2026 Conference Withdrawn Submission_

### Official Review · Reviewer_2zpx · 2025-10-20

**Soundness:** 2
**Presentation:** 2
**Contribution:** 2
**Rating:** 2
**Confidence:** 4

**Summary:**

This paper proposes ARCA (Adaptive Reasoning via Cognitive Allocation), a structured reasoning framework designed to balance accuracy and efficiency in large language model (LLM) reasoning.

**Strengths:**

1. The topic of research is promising;

2. Experimental results are good.

**Weaknesses:**

1. I don’t think this paper provides any meaningful insights. Specifically, in Section 3.1, the authors decompose the task into several sub-tasks, similar to [1]. In Section 3.2, the conceptual novelty over prior structured-reasoning methods such as Tree-of-Thought, Graph-of-Thought, or Meta-Reasoner remains somewhat unclear. The paper largely re-packages known search-and-selection patterns under new terminology;

2. Many key components (e.g., Phase Generator, Phase Classifier, Direction Generator, Borda-aggregated selector) are vaguely defined and appear heuristic. It is not obvious how they are implemented, trained, or verified to work autonomously;

3. The framework illustration (Figure 2) is quite difficult to interpret. The authors introduce many specific technical terms without providing sufficient explanations or context. Furthermore, using Sudoku as an illustrative example is not ideal, as it assumes the reader has prior knowledge of the game, which may not always be the case. Overall, while I can grasp the general idea of the framework, the detailed mechanisms remain vague and unclear.


[1] Least-to-Most Prompting Enables Complex Reasoning in Large Language Models

**Questions:**

The paper claims to introduce “cognitive resource allocation,” but this is essentially adaptive search-depth control. Please clarify how ARCA differs theoretically from existing meta-reasoning or dynamic-depth frameworks.

---

> ### Author Response · Authors · 2025-11-19
>
> **1. For W1 and Q1. This paper doesn't provide any meaningful insights. Specifically, in Section 3.1, the authors decompose the task into several sub-tasks, similar to [1]. In Section 3.2, the conceptual novelty over prior structured-reasoning methods such as Tree-of-Thought, Graph-of-Thought, or Meta-Reasoner remains somewhat unclear. The paper largely re-packages known search-and-selection patterns under new terminology;**
>
> The reviewer notes that our work “re-packages known search-and-selection patterns under new terminology.” We respectfully disagree. As stated in the Introduction, our goal is indeed to leverage search-and-selection–based reasoning patterns, but with a focus on achieving both accuracy and efficiency, which is an explicit trade-off that existing methods do not fully address. Prior structured-reasoning approaches such as Tree-of-Thought, Graph-of-Thought, and Meta-Reasoner primarily rely on breadth-first exploration of large thought trees/graphs. While this enhances robustness, it also incurs substantial computational cost and leads to excessive use of cognitive resources.
>
> In contrast, ARCA introduces a shift from breadth-first exploration to a goal-oriented decision-making strategy. ARCA’s hierarchical reasoning framework, coupled with dynamic cognitive-resource allocation, enables it to adapt the depth and granularity of reasoning to the task state, rather than expanding thought structures uniformly. This mechanism is absent in prior work and constitutes the core conceptual difference: ARCA does not merely explore more thoughts—it decides how much thinking is needed, and when, based on an explicit resource-allocation objective.
>
> Empirically, this distinction translates into a substantially improved accuracy–cost trade-off. As shown in Figure 3, ARCA consistently lies on the Pareto frontier relative to prior methods, achieving comparable or higher accuracy with significantly lower inference-time expenditure. This demonstrates that ARCA provides a meaningful advance beyond existing structured-reasoning frameworks rather than repackaging them.
>
> **2. For W2 and W3: The components and terminology are insufficiently explained, Figure 2 is hard to interpret, and the Sudoku example assumes prior knowledge, leaving the framework’s mechanisms unclear.**
>
> We appreciate the reviewers’ concerns regarding the clarity of several components (e.g., Phase Generator, Phase Classifier, Direction Generator, Borda-aggregated Selector) and the interpretability of Figure 2. We acknowledge that our current version does not provide a brief introduction to Sudoku, which may make the illustrative example harder to follow for readers unfamiliar with the game. We will add a concise explanation of Sudoku rules in the revision.
>
> Regarding the framework description, we believe Figure 2 outlines the full workflow of ARCA on the Sudoku example, but we agree that additional clarification would help. The detailed mechanisms of each component are in fact presented in Sections 3.1–3.3:
>
> -   **Section 3.1** formalizes the Reasoning Chain Construction process. The LLM decomposes the task into key steps in this phase.
>
> -   **Section 3.2** describes the Cognitive Resource Allocation module, which includes:
>     -   How the Phase Classifier determines the current reasoning phase.
>     -   How the Direction Generator produces candidate reasoning directions.
>     -   How ARCA identifies the most promising path.
>
> -   **Section 3.3** provides the algorithmic details of the Borda-aggregated direction selection. This is the core mechanism enabling ARCA to allocate cognitive resources efficiently.
>
> -   For additional clarity, Appendix C contains expanded algorithmic steps and example prompts.
>
> To better evaluate how these components function, we conducted additional experiments in Appendix B.3 of the current version, demonstrating that each component indeed contributes as intended. In the next version, we will add a brief explanation of Sudoku so that readers can better understand how ARCA operates, and we will further polish the descriptions to improve the overall clarity of the manuscript.

---

### Official Review · Reviewer_HZU1 · 2025-10-30

**Soundness:** 1
**Presentation:** 3
**Contribution:** 2
**Rating:** 4
**Confidence:** 4

**Summary:**

This paper proposes an Adaptive Reasoning via Cognitive Allocation framework under a carefully designed, prompt engineering flow with a robust scoring tool. The aim is to address the trade-off between accuracy and efficiency in LLM reasoning, achieving high accuracy while maintaining efficient reasoning. The core argument is that instead of letting a large model think blindly, it's better to let it intelligently plan its thinking process, allocating more computational resources only where necessary, thus avoiding performance degradation due to over-reasoning.
The ARCA framework has two core modules:
1) Reasoning Chain Construction: Using a Phase Generator to decompose complex tasks into logically ordered reasoning stages.
2) Cognitive Resource Allocation: Dynamically allocating reasoning resources at both the macro-level (Phase Classifier) ​​and micro-level (Direction Generator + Borda-Aggregated Selection).

The paper conducts experiments on six reasoning tasks, including AQUA, BBEH, GSM8K, Game of 24, Sudoku, and AIME, claiming superior accuracy and efficiency compared to existing baseline methods.

**Strengths:**

S1.The paper addresses a critical challenge in LLM reasoning: balancing accuracy and efficiency. This is highly relevant given the high inference costs of reasoning models. The motivation is clearly articulated through an intuitive Sudoku example (Figure 1), effectively demonstrating that some reasoning phases require minimal cognitive resources ("think less") while others demand deeper exploration ("think more"). The problem formulation of "cognitive resource allocation" provides a fresh perspective on reasoning efficiency.

S2.The two-stage ARCA framework is well-structured. Stage 1 (Reasoning Chain Construction) decomposes tasks into logical phases, providing structured guidance to prevent fragmented reasoning. Stage 2 (Cognitive Resource Allocation) operates at two complementary levels: macro-level phase identification and micro-level direction selection. This hierarchical design is conceptually sound.

S3.The experimental work is substantial, covering six diverse reasoning tasks spanning different domains.

S4.The paper is generally well-written with clear structure and logical flow

S5.The paper provides detailed implementation specifics in the appendices, including prompts for Phase Generator, Phase Classifier, and Direction Generator (Appendix C.2), as well as algorithmic details for Borda-aggregated selection (Algorithm 1). This level of detail facilitates reproducibility and practical adoption.

**Weaknesses:**

W1. The core of this paper is to use LLM to judge the merits of inference directions, but the reliability of this hypothesis has not been verified. A key missing experiment is: for tasks with ground truth, actually execute each candidate direction to see which one actually yields the correct answer, and then calculate the accuracy of the LLM+Borda selection.

W2. In multi-step inference, is there a possibility of the final step being a "lucky guess"? Does the planning in each phase truly play a role? The paper's logic is "use LLM to judge -> Borda aggregation -> select the optimal," but the definition of "optimal" comes from LLM itself, forming a circular argument and failing to distinguish whether the improvement truly finds a better path or reinforces model bias.

W3. Are there systematic biases in the underlying LLM system? For example, positional bias might not be detected: LLM might favor the first or last option. Style and phrasing biases could also exist: LLM might be drawn to certain "academic" terms. ARCA performs well on DeepSeek-V3 simply because it happens to cater to the specific biases and preferences of that model, rather than because the method itself is general.

W4. The paper claims that Borda is better, but lacks theoretical support. Key issue: The voting theory assumptions maybe not applicable. Borda theory is based on multiple independent rational voters, stable preferences, and no position effect, but LLM is a single model called multiple times, which may have systematic bias, random output, and obvious position bias. The paper directly borrows the conclusions but does not verify the rationality of the hypothesis transfer.

W5. ARCA requires multiple LLM calls (Phase Generator + Classifier + Direction Generator + Borda comparison), while CoT may only requires once. Although the number of tokens is similar, the latency cost of multiple rounds of interaction is negligible. Compare this to the accuracy of CoT with the same token budget.

W6. Only the Borda parameter was ablated, lacking ablation of Phase Generator, Phase Classifier, different aggregation methods, and different LLM evaluators, making it impossible to understand the true contribution of each component.

W7. The problems criticized in the paper may also apply to itself. (1) It criticizes the "structural rigidity" of Skills-in-Context, but ARCA's Phase Generator also predefines the phase structure, and if the LLM generation quality is poor (as the paper acknowledges), this "automatic rigidity" is even more dangerous; (2) It criticizes the "specialization" of Meta-Reasoner, but ARCA also needs to tune hyperparameters (m, n, max_directions, max_depth) for different tasks, and lacks a meta-learning framework; (3) It does not clearly distinguish itself from core works such as CToT and Self-Refine, and fails to verify the marginal contributions of Phase and Borda through ablation experiments. The paper should discuss the advantages and disadvantages of each method more objectively and in a balanced way.

**Questions:**

Q1. The paper acknowledges in Appendix B.3 that the generated phases are "generally broader and less consistent than those produced by human experts." Has anyone compared the performance differences between manually designed phases and automatically generated phases?

Q2. If the Phase Generator generates unreasonable phases (such as incorrect phase order), can subsequent Phase Classifiers and selections correct them?

Q3. Page 6 of the paper mentions using the Borda score instead of the Copeland score because of "robustness against minor inconsistencies." Could you provide theoretical analysis or experimental evidence to prove that the Borda score is indeed more robust than the Copeland score in your setting? A known problem with the Borda score is its susceptibility to irrelevant alternatives. Would this be a problem in your scenario?

Q4. Why does your method show significant improvement in some tasks but only slight improvement in others? Could you analyze which types of tasks are best suited to your method?

Q5.The main results of the paper are based on DeepSeek-V3, with results for Qwen3-8B in the appendix. How does it perform for smaller models (e.g., 7B, 1B)? Smaller models may not perform well in Phase Generation and Direction Generation. How does it perform for other architectures (e.g., Claude, GPT-4)? Have you tested the differences between open-source and pripretary models?

---

> ### Author Response · Authors · 2025-11-19
>
> **1. For W1, W2 and Q1. The paper's core premise lacks validation. Furthermore, the multi-step inference may involve lucky guesses, and the entire selection process is circular: the definition of an "optimal" path is determined by the LLM's own judgments, making it impossible to distinguish genuine improvement from the reinforcement of model bias.**
>
> Thank you for your feedback. To verify that my reasoning process is logical and not based on guessing the answer at the end, as well as to confirm the appropriateness of the Borda selection, we have provided an example of our complete reasoning process in Appendix C.3 in the rebuttal revision version. This includes the reasoning steps, phases, and Borda selection results. Although an expert evaluation model or phase list is not included for these tasks, our reasoning process aligns with human logic, with each step being reasonably sound, ensuring the overall accuracy of the process.
>
> **2. For W3, Q5. The method's generalizability is questionable due to potential systematic biases in the LLM, such as positional or stylistic preferences, which may mean ARCA's success on DeepSeek-V3 is not a result of its inherent strength but rather an alignment with that specific model's biases. The evaluation is also limited, as performance on smaller models and other architectures remains untested, leaving it unclear whether the approach is robust or simply overfitted to a single model.**
>
> To validate the generalization capability of our model, we have included experiments on smaller models (Qwen2.5 7B) and the closed-source model (GPT-4o mini) in Appendix B.2 of the rebuttal revision version. The results demonstrate that our method effectively adapts to different models, exhibiting strong generalization performance. Additionally, to address concerns regarding potential ordering bias, we have conducted further experiments detailed in Appendix B.3 of the rebuttal revision version. The experimental results indicate that our approach is largely unaffected by model position bias.
>
> **3. For W4, Q3. The paper directly borrows the conclusions but does not verify the rationality of the hypothesis transfer. Could you provide theoretical analysis or experimental evidence to prove that the Borda score is indeed more robust than the Copeland score in your setting?**
>
> Thank you for your comments regarding the performance of the Borda score in LLMs. To demonstrate the effectiveness of the Borda score in handling LLM preference feedback, we have included a reference experiment in Appendix C.4 of the rebuttal revision version that simulates LLM uncertainty and tie situations. The results show that the Borda score method outperforms the Copeland score method in simulated environments, demonstrating better capability in handling comparative ranking of LLM feedback. Furthermore, since each reasoning path is assigned a specific phase, the overall direction of reasoning is effectively constrained, minimizing the generation of irrelevant content.
>
> **4. For W5. ARCA requires multiple LLM calls, while CoT may only requires once. Although the number of tokens is similar, the latency cost of multiple rounds of interaction is negligible. Compare this to the accuracy of CoT with the same token budget.**
>
> As the number of model invocations increases, the associated cost will inevitably rise. However, our core motivation is to balance cost and performance, rather than comparing performance under equal cost conditions. Therefore, the results presented in Figure 3 represent the optimal trade-off between these two factors, rather than displaying cost per accuracy unit. How to reduce the number of model invocations remains an important direction for future work.

---

> ### Author Response · Authors · 2025-11-19
>
> **5. For W6, Q2 and Q4. Only the Borda parameter was ablated, lacking ablation of Phase Generator, Phase Classifier, different aggregation methods, and different LLM evaluators, making it impossible to understand the true contribution of each component. If the Phase Generator generates unreasonable phases, can subsequent Phase Classifiers and selections correct them? Why does your method show significant improvement in some tasks but only slight improvement in others? Could you analyze which types of tasks are best suited to your method?**
>
> To further analyze the contribution of each module in our model, we conducted experiments on the marginal contribution of individual modules in Appendix B.3 of the rebuttal revision version. The experimental results demonstrate that the phase module effectively guides our method in complex reasoning tasks, while the Borda selection module helps the model efficiently choose the optimal reasoning path at each step.
>
> Since the phases we generate are not sequential subtasks that must be completed one by one, but rather serve as macro-level directional guidance, the reasoning process does not necessarily need to go through all phases. Additionally, the classifier determines the most likely phase for the current step, avoiding obviously incorrect phases and effectively preventing misordering issues.
>
> Regarding the experimental results, as the difficulty varies across tasks, our method exhibits differentiated improvements. For simpler tasks like AQUA, where the baseline model already performs competently, our method further enhances performance while achieving a better cost-performance trade-off. For more complex tasks, such as BBEH, where the baseline model struggles to accomplish the task, our phase configuration and selection module enable the model to handle these challenges, resulting in more significant improvements compared to simpler tasks.
>
> **6. For W7. The paper should discuss the advantages and disadvantages of each method more objectively and in a balanced way.**
>
> Regarding Skills-in-Context methods, these approaches rely on predefined skill libraries, which significantly limit their generalization across different tasks. When no suitable skill is available, they can only approximate by sampling from the existing skill library. In contrast, ARCA generates task-specific phases that serve as macro-level guidance rather than detailed actions. These phases adapt automatically to different tasks, and as evidenced by our experimental results, the macro guidance combined with multi-path selection offers greater tolerance for potential generation errors.
>
> As for Meta Reasoner, this method requires predefined basic operations and the training of a contextual bandit. However, in the context of large language models, it is often challenging to train such a model to convergence quickly, leading to difficulties in task transfer. Our method, on the other hand, requires no pretraining and operates through real-time reasoning, offering greater flexibility. Additionally, as shown in our ablation study, the algorithm does not require highly fine-tuned parameters, and a relatively broad parameter range can yield satisfactory results, making it easier to deploy across various tasks.
>
> For pairwise comparison methods like CToT, the lack of guidance and constraints in generated nodes often necessitates retaining previously evaluated candidates and re-comparing them with new options in subsequent rounds. As the depth of reasoning increases, this leads to a significant computational overhead. In contrast, ARCA informs the reasoner of phase information to constrain the reasoning direction, while employing Borda selection and pruning strategies to reduce computational load and select optimal nodes efficiently, thereby substantially saving cognitive resources during reasoning.

---

### Official Review · Reviewer_b2YF · 2025-10-31

**Soundness:** 3
**Presentation:** 3
**Contribution:** 2
**Rating:** 6
**Confidence:** 4

**Summary:**

This article proposes the Adaptive Resource Allocation for Cognition (ARCA) framework to address the trade-off between accuracy and efficiency in Large Language Model reasoning, avoiding overthinking and resource waste caused by traditional COT approaches. The ARCA framework decomposes complex tasks into structured reasoning phases via ReasonGen and dynamically focuses resources using PhaseClass. At the micro level, it generates multiple reasoning directions and employs a Borda aggregation selection mechanism based on LLM preference feedback to identify the optimal reasoning path. Experimental results demonstrate that ARCA significantly improves accuracy while reducing reasoning costs, making it particularly suitable for complex long-chain reasoning tasks.

**Strengths:**

1.It presents a novel and timely perspective by addressing the efficiency-accuracy trade-off caused by "overthinking" in LLM reasoning. For the first time, it introduces a structured decomposition approach and a dynamic hierarchical resource allocation mechanism.
2.The paper features a clear line of reasoning, drawing inspiration from the hierarchical cognitive processes through which humans solve problems.

**Weaknesses:**

1. The effectiveness of ARCA depends on ReasonGen generating valid phase structures. While tasks like Sudoku and 24-point have clear structures, can ReasonGen introduce errors or miss key phases in more open-ended tasks requiring complex domain knowledge?
2.Given that all ARCA components (main solver, ReasonGen, PhaseClass, and LLM Evaluator) rely on a single large LLM (e.g., GPT-4), does this lead to prohibitively high inference costs?
3.ARCA claims efficiency, yet the Borda aggregation mechanism requires generating multiple candidates and performing $O(N^2)$ pairwise comparisons, which introduces additional inference calls and delays, conflicting with the claim of negligible overhead.

**Questions:**

What are the criteria for switching between inference stages (e.g., from "locating potential numbers" to "eliminating impossible options")? Do these transition conditions have soft or hard cutoffs? We are concerned that near the end of a stage, the LLM might experience a "transitional oscillation," repeatedly switching between adjacent stages and thereby wasting resources instead of conserving them.

---

> ### Author Response · Authors · 2025-11-19
>
> **1. Can ReasonGen introduce errors or miss key phases in more open-ended tasks requiring complex domain knowledge?**
>
> Thank you for your feedback. We acknowledge that this is indeed a challenging issue and represents an open problem: generating fully correct results in open-ended tasks with complex domain knowledge is inherently difficult. Our work focuses on improving the efficiency and performance of LLM decision-making within its capability range, rather than solving this broader open problem.
>
> The phases generated by our framework serve primarily to guide the reasoning process, as opposed to functioning as compulsory subtasks that must be completed sequentially to solve the problem. Accordingly, during reasoning, a classifier determines the most probable current phase, and the selection of an optimal path helps mitigate the impact of potential phase misidentification. Since these phases are not mandatory subtasks, the system is not required to traverse all of them, and the absence of some phases has minimal impact on the overall reasoning outcome.
>
> **2. Given that all ARCA components (main solver, ReasonGen, PhaseClass, and LLM Evaluator) rely on a single large LLM (e.g., GPT-4), does this lead to prohibitively high inference costs?**
>
> The core motivation of our work is precisely to navigate the trade-off between computational cost and reasoning accuracy. While the introduced modules do incur additional overhead, they correspondingly enhance performance. As demonstrated in Figure 3, our method is designed to actively seek and operate at the optimal balance point between resource consumption and performance gains, which is a deliberate design choice rather than an unintended cost.
>
> **3. ARCA claims efficiency, yet the Borda aggregation mechanism requires generating multiple candidates, which introduces additional inference calls and delays, conflicting with the claim of negligible overhead.**
>
> We have included a complexity analysis of the algorithm in Appendix C.1. The results demonstrate that the algorithm achieves a complexity of $\mathcal{O}(nK)$ under optimal conditions, degrading to $\mathcal{O}(nK^2)$ only in the worst case. Consequently, our method outperforms pairwise comparison in efficiency across most practical scenarios. By incorporating appropriate pruning strategies, the Borda selection algorithm effectively balances efficiency and performance while reducing computational costs.
>
> **4. What are the criteria for switching between inference stages (e.g., from "locating potential numbers" to "eliminating impossible options")? Do these transition conditions have soft or hard cutoffs?**
>
> In our framework, phase transitions are determined dynamically by a phase classifier. Specifically, based on the current reasoning state, the system first assesses whether the ongoing phase is complete. If so, it then identifies the most appropriate subsequent phase from the phase list. Our experimental results demonstrate that this switching mechanism operates accurately and reasonably on comprehensible tasks.
>
> **5. We are concerned that near the end of a stage, the LLM might experience a "transitional oscillation," repeatedly switching between adjacent stages and thereby wasting resources instead of conserving them.**
>
> Thank you for raising the point about potential repeated switching. Our phase classifier is designed with global awareness of the complete phase list, the current phase, and the history of previously completed phases. This grants the classifier a clear understanding of the overall reasoning progress, thereby enabling it to make stable transitions and effectively prevent oscillations, which maintains high efficiency. In the next version, we will provide a more detailed explanation to help readers understand this mechanism clearly.

---

### Official Review · Reviewer_taAD · 2025-10-31

**Soundness:** 2
**Presentation:** 2
**Contribution:** 2
**Rating:** 4
**Confidence:** 3

**Summary:**

To address the trade-off between accuracy and efficiency in long chain-of-thought reasoning, this paper proposes Adaptive Reasoning via Cognitive Allocation. By combining reasoning-chain construction and cognitive-resource allocation, the method guides structured reasoning and selects more promising reasoning paths. Experiments are conducted on a broad range of reasoning tasks.

**Strengths:**

The paper proposes an adaptive cognitive allocation mechanism that uses a macro/micro hierarchical structure combined with the Borda aggregation algorithm to evaluate different reasoning steps and prioritize the most promising ones. This provides a practical idea for pruning reasoning paths in multi-step reasoning.

**Weaknesses:**

The paper lacks clear novelty. The authors should better clarify how their method differs from Tree of Thoughts (ToT) and Graph of Thoughts (GoT), which also employ multi-step reasoning with verification. Although Borda scoring is used, it appears conceptually similar to Pairwise Comparison [1], differing mainly in the scoring formulation (win count vs. win rate).

Each stage of the proposed method is generated via prompting an LLM, meaning the performance is highly dependent on how well the LLM interprets and executes instructions. Prior research [2–3] shows that LLMs are sensitive to prompt positional information, which may affect reproducibility and stability.

In Section B.3, the authors state:”While these phases are often broader and exhibit less consistency compared to those created by human experts, they prove to be sufficiently accurate to effectively guide the reasoning process.”
It remains unclear how these phase divisions were evaluated against human-created ones — what metrics or criteria were used?

[1]Zhang Z Y, Han S, Yao H, et al. Generating Chain-of-Thoughts with a Pairwise-Comparison Approach to Searching for the Most Promising Intermediate Thought[C]//International Conference on Machine Learning. PMLR, 2024: 58967-58983.

[2]Guo Y, Guo M, Su J, et al. Bias in large language models: Origin, evaluation, and mitigation[J]. arXiv preprint arXiv:2411.10915, 2024.

[3]Zheng C, Zhou H, Meng F, et al. Large language models are not robust multiple choice selectors[J]. arXiv preprint arXiv:2309.03882, 2023.

**Questions:**

1.Since each stage of the method is generated via prompting, I suggest performing LLM consistency evaluation by executing the same prompts multiple times (and swapping candidate order) to test stability.

2.Compare the generated reasoning paths with human reasoning trajectories on the same tasks to analyze the method’s advantages or differences.

3.The evaluation of candidate reasoning steps is relatively simple; how efficient is Borda aggregation when applied to large-scale or more complex reasoning paths?

4.How is the reasoning cost specifically computed?

---

> ### Author Response · Authors · 2025-11-19
>
> **1. The authors should better clarify how their method differs from ToT and GoT. It appears conceptually similar to Pairwise Comparison, differing mainly in the scoring formulation.**
>
> ARCA differs fundamentally from Tree of Thoughts or Graph of Thoughts in its core paradigm: it shifts from a breadth-first search approach to a phase-oriented decision-making strategy. While ToT and GoT enhance robustness by constructing and traversing extensive thought trees or graphs, their methods are computationally expensive and often lead to excessive consumption of cognitive resources. In contrast, ARCA employs a hierarchical reasoning framework with a dynamic cognitive resource allocation mechanism, allowing it to maintain high accuracy while significantly improving efficiency.
>
> The key advance of the ARCA framework beyond the pairwise comparison paradigm is evident in two aspects. In terms of evaluation logic, ARCA uses a Borda score based on average win rates. This global averaging mechanism smooths out stochastic noise in LLM outputs, delivering more robust results than the Copeland score which relies on single duel outcomes, while also being more resource-efficient than methods requiring multiple pairwise comparisons, such as CToT[1]. Regarding resource strategy, ARCA introduces dynamic pruning—first rapidly eliminating inferior options, then focusing precise evaluation on a small set of high-quality candidates. By optimizing the number of comparisons, ARCA achieves a better balance between accuracy and computational efficiency.
>
> [1] Generating chain-of-thoughts with a pairwise-comparison approach to searching for the most promising intermediate thought.(Zhang et, al. 2024)
>
> **2. Prior research [2–3] shows that LLMs are sensitive to prompt positional information, which may affect reproducibility and stability.**
>
> Thank you for your valuable feedback. To address the concerns regarding potential ordering bias, we have conducted additional experiments detailed in Appendix B.3 in the rebuttal revision version. In these randomized trials, we shuffled the candidate options prior to each selection round to eliminate any positional bias. The results confirm the overall stability and demonstrated robustness of our selection method, showing that it is unaffected by the specific order of candidates. Additionally, all results presented in the main text represent the average of three independent trials, which effectively ensures stability against prompt variations.
>
> **3. It remains unclear how these phase divisions were evaluated against human-created ones — what metrics or criteria were used?**
>
> It is important to note that our generated phase example​ is provided solely to help readers identify the results of our method. To compare the phases generated by our method against human-constructed ones, we employed a qualitative analysis rather than a specific metric. This analysis assessed how well the phases reflect the logical steps and reasoning processes humans use to solve the tasks. This approach was necessitated by the lack of a dedicated dataset of expert thought processes. Our evaluation shows that the generated phases are sufficiently accurate to effectively guide the reasoning process.
>
> **4. Compare the generated reasoning paths with human reasoning trajectories on the same tasks to analyze the method’s advantages or differences.**
>
> A comparative analysis of our reasoning process is provided in Appendix C.3 in the rebuttal revision version, which elaborates on the application of our method across various tasks. The documented reasoning paths and selection decisions demonstrate that the overall workflow and the final choices align closely with human logic, thereby validating the method's soundness.
>
> **5. The evaluation of candidate reasoning steps is relatively simple; how efficient is Borda aggregation when applied to large-scale or more complex reasoning paths?**
>
> We provide a formal complexity analysis in Appendix C.1, showing that our algorithm achieves $\mathcal{O}(nK)$ complexity in typical cases, with worst-case performance bounded by $\mathcal{O}(nK^2)$. This scalability advantage becomes particularly significant in large-scale tasks, where our method consistently outperforms pairwise comparison, whose quadratic complexity $\mathcal{O}(K^2)$ becomes prohibitive as $K$ grows. Through adaptive pruning, the Borda selection mechanism maintains a favorable trade-off between accuracy and efficiency, enabling practical deployment even as the candidate pool expands.
>
> **6. How is the reasoning cost specifically computed?**
>
> The computational cost in our approach is calculated with reference to the formula used in CToT. The total cost is jointly determined by the number of completion tokens and prompt tokens.

---

### Note · Authors · 2026-01-03

I have read and agree with the venue's withdrawal policy on behalf of myself and my co-authors.